# Effects of Hormonal Profile, Weight, and Body Image on Sexual Function in Women with Polycystic Ovary Syndrome

**DOI:** 10.3390/healthcare11101488

**Published:** 2023-05-19

**Authors:** Ana-Maria Cristina Daescu, Liana Dehelean, Dan-Bogdan Navolan, Alexandru-Ioan Gaitoane, Andrei Daescu, Dana Stoian

**Affiliations:** 1Doctoral School Department, Victor Babes University of Medicine and Pharmacy, 300041 Timisoara, Romania; ana-maria.daescu@umft.ro; 2Department of Internal Medicine II, Victor Babes University of Medicine and Pharmacy, 300041 Timisoara, Romania; stoian.dana@umft.ro; 3Neurosciences Department, Victor Babes University of Medicine and Pharmacy, 300041 Timisoara, Romania; 4Department of Obstetrics and Gynecology, Victor Babes University of Medicine and Pharmacy, 300041 Timisoara, Romania; navolan.dan@umft.ro; 5Department of Mathematics, Politehnica University of Timisoara, 300006 Timisoara, Romania; alexandru.gaitoane@student.upt.ro (A.-I.G.); andrei.daescu2@student.upt.ro (A.D.)

**Keywords:** PCOS, female sexual dysfunction, FSFI, BESAQ

## Abstract

Polycystic ovary syndrome (PCOS) is a hyperandrogenic endocrinological disorder associated with chronic oligo-anovulation and polycystic ovarian morphology. Compared to women without PCOS, women with PCOS have a risk of sexual dysfunction that is more than 30% higher. Although alterations in sex hormones and psychosocial wellbeing have been proposed, the precise mechanisms of FSD in PCOS remain unclear. The aim of our study was to analyze how the hormonal, clinical and psychometric parameters of PCOS patients are involved in the development of sexual dysfunction. The study group consisted of 54 women, aged between 21 and 32 years, diagnosed with PCOS. We collected the following parameters: age, body mass index (BMI), the Ferriman–Gallwey score (FG), maximum duration of oligomenorrhea, abdominal circumference (AC), free testosterone value (FT), luteinizing hormone/follicle stimulating hormone (LH/FSH) ratio value, serum cortisol value and ovarian ultrasound appearance. At the time of the examination, patients were asked to fill in the Female Sexual Function Index (FSFI) and the Body Exposure during Sexual Activities Questionnaire (BESAQ). Statistically significant differences were observed between normal weight and overweight women regarding BESAQ (*p*-value = 0.02) and FSFI total (*p*-value <0.001). Elevated BMI, AC or BESAQ scores correlated with a lower FSFI score. The most involved domains of the scale were orgasm, arousal, and desire. Elevated BESAQ scores increase the risk of female sexual dysfunction (FSD) by 4.24 times. FT, BESAQ score, BMI, and LH/FSH ratio were found to independently predict FSD. The cutoff point for the BESAQ score in detecting FSD was found to be 1.97. Weight, body image and anxiety related to sexual activities seem to be significant components in the development of sexual dysfunction in PCOS patients, beyond the effect due to hyperandrogenism. FT value has a U-shape effect in sexual dysfunction, because both in the case of deficit and in the case of excess, sexual function is impaired. BESAQ is a strong predictor for sexual dysfunction in women with PCOS, along with FT value, LH/FSH ratio and BMI.

## 1. Introduction

Polycystic ovary syndrome (PCOS) is a hyperandrogenic endocrinological disorder associated with chronic oligo-anovulation and polycystic ovarian morphology. It is often associated with psychological issues, such as body image disturbance; psychiatric disorders, including depression and anxiety; metabolic disorders, namely insulin resistance (IR) and compensatory hyperinsulinemia (HI), recognized as a major factor responsible for altering metabolism and androgen production [1,2,3]. Most women with PCOS are overweight or obese, thus increasing the secretion of androgens, affecting metabolism and reproductive functions, and possibly promoting the development of the clinical phenotype of PCOS [3].

The estimated prevalence of PCOS indicates that it is a common endocrinological condition in the population, affecting 4–8% of women of reproductive age. However, it has been shown that the prevalence differs depending on the diagnostic criteria used, so studies show that the prevalence after using the Rotterdam criteria is twice as high as using the Androgen Excess and PCOS Society criteria, and even three times higher compared to using the criteria of the National Institute of Health [4,5,6,7]. The family history of PCOS is a risk factor for it. Given the family conglomeration of cases, PCOS is considered a hereditary disease. A high prevalence of PCOS among first-degree relatives is suggestive for a genetic influence [8].

The increased prevalence of PCOS is associated with a number of conditions. A history of weight gain usually precedes the development of the clinical features of PCOS, and it has been shown that adopting a healthy lifestyle leads to a reduction in body weight, abdominal circumference, testosterone, insulin-resistance and hirsutism in women with PCOS [9]. Several studies indicate that while obesity may increase the risk of developing PCOS, the prevalence rates of PCOS in underweight, normal-weight, and overweight women with varying degrees of obesity are all relatively similar. Specifically, the prevalence rates for underweight, normal-weight and overweight women are approximately 8.2%, 9.8%, and 9.9% respectively. For overweight women with mild, moderate, and severe obesity the prevalence rates are 5.2%, 12.4%, and 11.4%, respectively. Therefore, the authors concluded that obesity may have a modest effect on the risk of developing PCOS [10]. The increased prevalence of PCOS has also been associated with type 1, type 2 and gestational diabetes. Studies show that the prevalence of PCOS is 40.5% in women with type 1 diabetes, 82% in women with type 2 diabetes and 16% in women with gestational diabetes [11]. A number of factors associated with a high risk of developing PCOS have been identified in the female pediatric population. Thus, an increased prevalence is associated with prenatal factors, such as high birth weight of girls born to overweight mothers and low birth weight. Early puberty, obesity and metabolic syndrome in children, as well as irregular menstruation in adolescents, are also associated with an increased prevalence [12].

The clinical elements of PCOS are divided into somatic symptoms caused by hyperandrogenism (acne, seborrhea, hirsutism), menstrual symptoms (oligomenorrhea or amenorrhea) with subsequent infertility, and associated clinical phenomena (dysmenorrhea, premenstrual syndrome, obesity, metabolic syndrome), having a significant psychological morbidity with a negative effect on quality of life [8].

In 1990, the National Institute of Health developed a diagnostic algorithm for PCOS, according to which patients could be diagnosed only on the basis of the presence of menstrual disorders and hyperandrogenism (HA), while excluding other etiologies [13]. The European Society for Reproduction and Endocrinology and the American Society of Reproductive Medicine drafted the “Rotterdam criteria” in 2003, according to which, PCOS is defined by the presence of two of the following three signs: oligo-ovulation or anovulation, HA and ultrasound-detected polycystic ovaries. HA can be clinical or biochemical or both, and polycystic ovaries are considered ovaries with 12 or more immature follicles with a diameter of 2–9 mm. The Rotterdam criteria applies only in the context of the exclusion of other causes. They have now been adopted worldwide and are used in most studies and publications [8]. In 2006, the Androgen Excess and PCOS Society argued that only two criteria were needed, namely HA and ovarian dysfunction, of course to the exclusion of other causes. HA can be biochemical or clinical (hirsutism, alopecia, acne) or both, and ovarian dysfunction involves oligo-ovulation or anovulation, polycystic ovaries or both [8].

Because a root cause has not yet been elucidated, the most accepted premise is that of a multifactorial cause, where the interactions between environmental factors and individual intrinsic factors act synergistically, leading to a common result translated by HA, the biochemical marker of this pathology [14]. In women with PCOS, elevated anti-mullerian hormone (AMH) levels play an important role in disrupting long-term ovarian physiology with high concentrations being consistent with an increased rate of infertility [15,16,17,18]. Feedback disorders in the hypothalamic-pituitary-ovarian axis are another typical feature of PCOS with increased an frequency and amplitude of gonadotropin-releasing hormone (GnRH) secretion and pulsatile secretion of luteinizing hormone (LH). Elevated levels of this hormone induce increased androgen synthesis by ovarian thecal cells. In turn, HA induces a decrease in the sensitivity of the feedback to the gonadotropic hypothalamic cells, strengthening the hypersecretion of GnRH and LH. This is one of the pathological mechanisms in which HA plays a pivotal role in the emergence and progress of PCOS [19,20]. Genetic factors are also considered to play an important role in the development of the syndrome, being determinants of the environment conducive to abnormal androgen synthesis in ovarian tissue. Mutations in androgen receptor genes, sex hormone binding globulin (SHBG), and numerous steroidogenic enzymes may be of major importance in predisposing the development of PCOS [16].

Patients with PCOS are more susceptible to mental health issues. Patients with PCOS experience physical and metabolic abnormalities that directly contribute to psychological issues like depression, anxiety, eating and sexual disorders [21]. Previous studies aimed to investigate the prevalence of depression, anxiety, and perceived stress in women with PCOS. Results showed that women with PCOS had a higher prevalence of depression, anxiety and perceived stress compared to women without PCOS [22]. PCOS patients presented indicators of exacerbated psychological discomfort that were found to be potentially contributing in the association between PCOS, depression and anxiety [23]. In addition, hirsutism, acne, alopecia, and other hyperandrogenic symptoms often make patients feel as though they have lost their attractiveness, which lowers their self-esteem and physical fulfillment and has a severe impact on their mental health [24]. Obesity and history of infertility were also found to be significant predictors of depression, anxiety, and lower sexual function in women with PCOS [22,25].

Compared to women without PCOS, women with PCOS have a risk of sexual dysfunction that is more than 30% higher [26]. Although alterations in sex hormones and psychosocial wellbeing have been proposed, the precise mechanisms of FSD in PCOS remain unclear. Low self-esteem, anxiety and depression can be brought on by menstrual irregularities. Additionally, obese and androgen-excessive women with PCOS may feel less attractive and may have difficulties regarding intimate relationships. [27]. There are a lot of studies that have focused on sexual function in patients with PCOS. In terms of sexual function, research has shown that PCOS patients can experience problems such as decreased sexual desire, difficulty reaching orgasm, and pain during intercourse [28]. The studies that used FSFI to assess sexual function in women with PCOS have found that PCOS patients have a lower mean score than healthy women, indicating lower sexual function. These problems can be caused by hormonal imbalance, high androgen levels, and other health problems associated with PCOS [29]. Regarding the BESAQ score, there are fewer studies that have used this score in the evaluation of sexuality in women with PCOS, even though both the FSFI and BESAQ are useful tools in the assessment of sexual function and can help identify sexual health problems and plan appropriate treatment [30,31,32]. High levels of serum testosterone, LH and an elevated LH/FSH ratio are all indicators of PCOS and despite the fact that previous studies shown that androgen insufficiency is associated with decreased sexual drive, research on women with PCOS is limited and has mixed findings [19,33].

The association between PCOS and female sexual dysfunction is not fully explained by body mass index and hormonal factors, indicating that PCOS itself may be a contributing factor to FSD [34]. The primary objective of our study was to investigate the role of hormonal, clinical, and psychological factors in the pathogenesis of sexual dysfunction in women with PCOS. Specifically, we aimed to analyze the relationship between sexual function, hormonal profile, weight status, and body image perception. By examining these factors, we aimed to provide a more comprehensive understanding of the mechanisms underlying sexual dysfunction in PCOS patients. Therefore, we considered our research useful in adding value to this field of study and in highlighting the need for a more comprehensive approach to PCOS management, including mental health support and sexual counseling.

## 2. Materials and Methods

### 2.1. Establishment of the Study Group

A total of 54 cases of patients above the age of 18, who meet at least two of the three Rotterdam criteria (oligo-ovulation or anovulation, HA and ultrasound-detected polycystic ovaries), engaged in sexual activity for at least four weeks before the study and had not received any form of treatment beforehand, were included in the study. Patients were recruited from the SCJUPBT Outpatient Endocrinology Clinic in Timisoara, Romania.

Other causes of similar clinical or paraclinical manifestations were excluded, namely: thyroid dysfunction (hypothyroidism, hyperthyroidism), 21-hydroxylase deficiency, congenital adrenal hyperplasia with delayed onset, ovarian stromal hyperplasia or syndrome, Cushing syndrome, familial hirsutism, virilizing tumors of the ovary or adrenal gland, multicystic ovaries, pituitary hyperprolactinemia (prolactinoma) or use of drugs with a virilizing effect. Additionally, illiterate women, women with neurological, psychiatric disorders and severe somatic disorders were excluded from this study.

To establish the database, we collected the following: age, body mass index, the Ferriman–Gallwey score, maximum duration of oligomenorrhea (in days), abdominal circumference, free testosterone value (FT), luteinizing hormone/follicle stimulating hormone (LH/FSH) ratio value, serum cortisol value and ovarian ultrasound appearance. Other parameters were evaluated to exclude other pathologies: total testosterone, SHGB, AMH, salivary cortisol, urinary cortisol and prolactin.

The study was performed in accordance with the Ethical Guidelines of the Helsinki Declaration and was approved by the Ethics Committee of Victor Babes University of Medicine and Pharmacy, Timisoara, Romania. All subjects agreed to the evaluation and provided their written informed consent prior to inclusion.

### 2.2. PCOS Diagnosis

#### 2.2.1. Ultrasound Diagnosis

Ultrasound diagnosis of PCOS involves the presence of ≥12 ovarian follicles with a diameter between 2–9 mm and/or an ovarian volume ≥10 cm^3^. This ultrasound image detected at the level of a single gonad is sufficient for diagnosis, while excluding other causes. There are some standard recommendations regarding the examination, namely the need to use two plans in the ultrasound determination, as well as measuring the follicles ≥10 mm in the longitudinal plan for the most accurate results. Three-dimensional ultrasound is used to determine ovarian volume, but the value of the sample has low sensitivity compared to the evaluation of ovarian follicles being reserved for particular cases where transvaginal ultrasound cannot be performed, or where visualization of the ovaries is difficult [35].

#### 2.2.2. Hormonal Assays

The LH/FSH ratio was analyzed by the immunochemical method with electrochemiluminescent detection (ECLIA). Free testosterone serum levels and serum cortisol levels were analyzed by the enzyme linked immunosorbent assay (ELISA). Venous blood samples were collected in the morning after a fasting interval of a minimum of 8 to 12 h, at least 2 h after the time of awakening. Blood specimens were collected between 7–10 a.m. with a minimum rest of 30 min before sampling. Patients were instructed to avoid caffeine-based drinks, smoking, stress and exercise before blood sampling. For free testosterone, a value of 0.84–3.4 pg/mL was considered normal. For serum cortisol, a value between 6–23 mcg/dL was considered normal. An LH/FSH ratio > 2 with an elevated baseline LH level in a patient with chronic anovulation was considered suggestive for PCOS.

#### 2.2.3. Clinical Evaluation

##### The Ferriman–Gallwey Score

The Ferriman–Gallwey score (FG) is a method that evaluates and quantifies hirsutism in women. The original method dates from 1961 and is based on assessing the presence of hair in 11 areas of the body. Subsequently, the score was modified, limited to the use of nine body areas (upper lip, chin, chest, upper back, lower back, upper abdomen, lower abdomen, arms, forearms—excluded in modified score; thighs, legs—excluded in the modified score). In the modified FG score (mFG), hair growth is assessed from 0 (absent) to 4 (extensive growth) in each of the nine locations. Therefore, the minimum score for a patient is 0, and the maximum score is 36. When interpreting the score, the ethnic group to which the patient belongs to should be taken into account, and the pilosity associated with each group. Thus, for a Caucasian woman, an FG score ≥8 is considered suggestive for HA [36].

##### Number of Days of Oligomenorrhea

Oligomenorrhea is defined as a decrease in the normal frequency of menstruation. Specifically, menstruation occurs at intervals of more than 35 days, four to nine times a year. Among women with PCOS, 80% have clinically manifested oligomenorrhea. Of these, those with menstrual cycles of more than 35 days show more pronounced signs of HA, and those with menstrual cycles of more than 3 months have the highest rates of IR and HA [37].

##### BMI

The body mass index is an efficient statistical method for assessing the weight of patients. Given that HA and IR in PCOS can cause obesity, this statistical index can determine the impact of PCOS on the patient’s weight. A BMI is considered to be a health risk if ≥25 [38].

### 2.3. Administration of Psychometric Questionnaires

At the time of the examination, patients were asked to fill in the Female Sexual Function Index and the Body Exposure during the Sexual Activities Questionnaire.

The Female Sexual Function Index (FSFI) is a scale for measuring female sexual function that includes 19 self-reporting themes and assigns scores for various aspects of a woman’s overall sexual function (desire, arousal, lubrication, orgasm, satisfaction and pain) [39]. The FSFI is undoubtedly a highly qualified and scientific inventory due to the availability of an exceptionally wide database of published studies, including results from numerous observational and interventional investigations in different study populations [40]. The FSFI has been translated into Romanian and validated, the FSFI-RO showing strong psychometric proprieties, similar to those of the original English version. A total FSFI score of 26.55 was identified as the threshold for differentiating those with and without sexual dysfunction [40].

The Body Exposure during Sexual Activities Questionnaire (BESAQ) is a scientifically validated self-report assessment of body-image experiences in the context of sexual relations. The BESAQ measures the extent to which, during sex, a person experiences a self-conscious or anxious focus on their body’s appearance and report desires/attempts to avoid the exposure of certain aspects of their body to sexual partners [41]. The BESAQ was translated into Romanian by one of the authors and then reverse translated. A team of experts in the field of sexual medicine approved of the final translation and a group of volunteers participated in an evaluation of the test–retest reliability of the Romanian version. A lower BESAQ score is associated with better body image during sexual activity. Higher scores have been found to be correlated with sexual dysfunction [42].

### 2.4. Statistical Analysis

Continuous variables, depending on distribution type, were presented as mean and standard deviation (SD) for normally distributed data (Gaussian), and as median and interquartile range (Q25–Q75) in the case of non-normally distributed data. The categorical variables were presented as frequency and percentages.

In order to summarize the characteristics of the undergoing study population, descriptive and inferential statistical analysis was performed. Assessing the distribution of continuous variables was done by using the Shapiro–Wilk test. To test the differences between variables, the Mann–Whitney U test was performed. To assess the effect size of the difference between variables, the biserial correlation coefficient was used. To determine the relationship between variables, correlation analysis was conducted. To discover the independent predictors and risk factors of female sexual dysfunction, we used multivariate linear and logistic regression. Building the model was done by the backward elimination method and choosing the best model involved using the Akaike information criterion (AIC). To evaluate the performance of the binary classification model and to select the optimal threshold, based on the trade-off between sensitivity and specificity, the receiver operating characteristic (ROC) curve and Youden Index was used.

Data were collected, processed, and analyzed using R programming language version 4.2.2 (31 October 2022 ucrt), and the results were presented in tabular and graphic form. A *p*-value of < 0.05 was considered to indicate a statistically significant difference, with a 95% confidence interval.

## 3. Results

### 3.1. Characteristics of the Study Group

The study group consists of 54 women aged between 21 and 32 years old diagnosed with PCOS. A large part of the sample comes from the urban environment (77.8%), most of them pursue higher education (74.1%) and at the time of the study, the majority are employed (75.9%); more than half of the women surveyed are married (55.6%) and also more than a half fit into the female sexual dysfunction criteria (59.3%). Regarding religion, the majority are Orthodox Christians (72.2%). The baseline characteristics of our study group are detailed below in Table 1, Table 2 and Table 3. 

### 3.2. Comparison between Normal Weight and Overweight Women Regarding Other Determinants

The Shapiro–Wilk test was used to assess the normality of the data, resulting in a non-Gaussian distribution. We considered the following thresholds to define body mass categories: 18.5–24.9 for normal weight and ≥25 for overweight. Out of the 54 patients included in the study, 19 were in the normal weight range, while 35 patients were overweight or obese. The differences between the normal weight and overweight women were highlighted using the Mann–Whitney U test, as shown in Table 4.

Statistically significant differences were observed between normal weight and overweight women regarding BESAQ (*p*-value = 0.02) and FSFI total (*p*-value < 0.001). There is no statistically significant difference between normal weight and overweight women when it comes to FT and LH/FSH ratio.

### 3.3. Correlation Analysis between FSFI Scores and Other Determinants

We assessed the normality of the distributions using the Shapiro–Wilk test, resulting in only a part of the variables meeting the normality criteria (*p*-value > 0.05): age, FG score, abdominal circumference, BESAQ and FSFI total. To determine the relationship between variables, we used the Pearson product–moment correlation coefficient for normally distributed data and the Spearman’s rank correlation coefficient in the case of non-normally distributions. We used Cohen’s convention to interpret the effect size, where a correlation coefficient of 0.50 or larger is thought to represent a strong correlation and then we tested the statistical significance of the relationship with a 95% confidence interval. The results of the correlation analysis are shown in Table 5.

We noticed that all the relationships between the variables are inversely proportional and, in most cases, elevated BMI, abdominal circumference or BESAQ scores correlate with a lower FSFI score. The most involved domains of the scale are orgasm, arousal, and desire. The FSFI is highly correlated with BESAQ (Figure 1), many important associations occurring between them. The most notable ones are with the arousal domain, as being the strongest (rho = −0.80, *p*-value < 0.001), and with the orgasm domain. The FSFI orgasm domain also has the most correlations to the other studied components (BMI, cortisol level, abdominal circumference, BESAQ).

### 3.4. Independent Risk Factors for the Diagnosis of Female Sexual Dysfunction

To identify the independent risk factors for the diagnosis of FSD, we used multivariate logistic regression. Initially, the model had several independent variables: age, BMI, OM, FT, LH/FSH, cortisol, FG score, AC and BESAQ score in relation to the dichotomous dependent variable (FSFI total score ≤26.55 being representative for positive FSD diagnosis and FSFI total >26.55 for negative FSD diagnosis). The backward elimination method was used to build the model, and Akaike information criteria (AIC) was applied to choose the best model. The results are displayed in Table 6.

The risk of developing FSD increase with a higher BESAQ score. Elevated BESAQ scores increase the risk of female sexual dysfunction by 4.24 times. This regression model explains 29.5% ($R^{2}$ = 0.295) of the diagnosis for female sexual dysfunction.

### 3.5. Independent Predictors for Female Sexual Dysfunction

Independent predictors of FSD were highlighted using multivariate linear regression. The dependent variable is represented by the FSFI total score, and the independent variables included in the regression analysis were age, BMI, OM, FT, LH/FSH, cortisol, FG score, AC and BESAQ score. The backward elimination method was used to build the model, and Akaike information criteria (AIC) was applied to choose the best model. The results are shown in Table 7.

FT, BESAQ score, BMI, and LH/FSH ratio were found to independently predict FSD. $R^{2}$ = 0.614 (61.4% variations in FSFI score are explained by the model). The increase in FT, BESAQ score, and BMI leads to a decrease of the total FSFI score, the relationship being inversely proportional and the increase in the LH/FSH ratio leads to an increase in FSFI total score, the relationship being directly proportional. The increase in FT, BESAQ score, and BMI by 1 unit results in a decrease in the FSFI total score by 2.24, 2.26 and 0.27, respectively. The increase in LH/FSH ratio by 1 unit results in an increase of the FSFI total score by 3.92.

### 3.6. Threshold Value for BESAQ Score Providing Positive Diagnosis for FSD

ROC curve statistics were used to determine the threshold value of the BESAQ score for the diagnosis of FSD. The cutoff point determined using the Youden index to diagnose FSD is 1.97 for the BESAQ score. Values greater than the threshold have a positive, predictive value for the diagnosis of FSD, and values below the threshold have a negative predictive value for the diagnosis. AUROC statistics are presented in Table 8 and Figure 2.

## 4. Discussion

One of the aims of our study was to analyze if there is a significant difference in terms of the patient’s hormonal profile and sexual function between normal weight and overweight patients. We found that although the hormonal background does not differ, there is still an impairment of sexual function and body image perception in overweight patients. Therefore, body weight impacts sexual function, probably through the perception of body image.

Another objective was to analyze the correlations between sexual function in PCOS patients and the hormonal profile, the clinical and psychometric indicators that we used. We found that overall, the study group was characterized by hyperandrogenism, more precisely the patients presented higher values of FT than the normal range. In the literature, testosterone supplementation has been associated with increased libido, increased energy levels and has been shown to improve mood and general well-being [43,44]. However, in our sample, the field of sexual desire was the most affected, according to the FSFI lower scores in this domain. Although our study focused specifically on the relationship between hyperandrogenism and sexual function in women with PCOS, previous research has suggested that testosterone deficiency can have negative effects on sexual function. Studies in the field have shown that women with androgen deficiency also show an impairment of sexual function [45,46]. Therefore, we observe a U-shaped effect of testosterone, because both testosterone deficiency and excess, as in the case of PCOS, affect sexual function. The patients included in the study also presented higher values of the LH/FSH ratio than those considered normal. These results mean that PCOS patients present impairments in the functioning of the gonadostat and are correlated with FSD also through a testosterone-mediated mechanism.

In regard to psychometric evaluation, results in our study showed that self-esteem and perception of body image in the context of sexual activities was measured by the BESAQ score, which globally influences sexual function. Impairments in this area might lead, through cognitive interference during sexual activities, to a negative body image that can affect orgasm and sexual desire. Our results confirm the previous hypotheses related to the fact that high BESAQ scores are associated with the impairment of sexual function and that women with more positive sexual self-schemas, who reported less avoidance and body-image anxiety, have a better sexual function [42]. Additionally, these women reported less frequent sexual desire and impaired orgasm.

Because obesity is frequently associated with low self-esteem and more anxious body-related behavior during sexual activities, it was expected that our results show that overweight and obese women show an impairment of sexual function in general, and of arousal and orgasm in particular [42,47]. A possible explanation might be that, even though increased weight is linked to an increase in estrogen, women’s sexual desire and psychological arousal are typically more strongly influenced by contextual factors and a person’s ability to evaluate relevant sexual stimuli on a cognitive–emotional level, rather than by biological factors that frequently play a supporting role [47,48,49]. Orgasm and body mass were also correlated in other previous studies that showed the benefits of weight loss in relation to orgasm frequency and quality [50].

We tested the diagnostic performance of the BESAQ for detecting FSD. In our sample, the results show a strong ability of the BESAQ to distinguish between patients with and without FSD. This finding might suggest that the perception of body image in a sexual context has a notable role in the development of FSD. Additional studies in this direction are needed.

Previous research examining the relationship between FSFI and BESAQ scores found conflicting results. While some studies suggested no correlation between the two, indicating no connection between body image perception and sexual function, others demonstrated a negative association between high BESAQ scores and low FSFI scores [51,52,53]. Our findings align with the latter group. Furthermore, our study showed that higher BESAQ scores can increase the risk of FSD and can even predict it. In addition, our results show a strong correlation between BESAQ scores and the arousal and orgasm domains of the FSFI. This may be due to a significant involvement of psychological factors in the female sexual response. Therefore, anxiety related to exposure of the body during sexual activity or low self-esteem may impact sexual function in women. This highlights the need for careful evaluation of the psychological component of sexuality and of the psychiatric symptoms linked to sexual contexts, such as anxiety and avoidance behavior during sexual activities. FT, the LH/FSH ratio and the BMI were also found to be strong predictors for FSD.

Our study is not without limitations. One of the limitations of our study is that we did not collect data on mental wellbeing, such as depression and anxiety, which may also have a significant impact on sexual function in women with PCOS. However, we did include the BESAQ to assess body image perception in women with PCOS, which has been shown to be significantly impacted by excess weight, anxiety, and avoidance behavior related to sexual activities. The small number of patients included in the study is motivated by the fact that we included only patients diagnosed with PCOS that met at least two out of three Rotterdam criteria, and who did not follow any kind of treatment before participating in the study, precisely in order not to influence the results. Another limitation is the fact that we did not include a control group and we did not follow the evolution of changes in sexual function after the initiation of treatment and further research is needed in this regard.

## 5. Conclusions

Our findings can explain the process by which the sexual function of women with PCOS is affected, despite hyperandrogenism, which in the literature has been associated with a better sexual function, especially from the point of view of libido. Excess weight leads to an impairment in body image perception, to anxiety and avoidance behavior related to sexual activities. These aspects seem to be a significant component of sexual dysfunction in the case of these patients, beyond the effect due to hormonal changes. This underscores the importance of addressing body image issues and anxiety related to sexual activities in the treatment of patients with PCOS.

FT value has a U-shape effect in terms of affecting female sexual function, because both in the case of deficit and in the case of excess, sexual function is impaired. This has important implications for the diagnosis and treatment of sexual dysfunction in women with PCOS.

BESAQ could be a useful tool in the diagnosis of sexual dysfunction in women with PCOS and the scale score is a strong predictor for sexual dysfunction, along with FT value, LH/FSH ratio and BMI. The scale score can help healthcare professionals identify patients who may be at risk for sexual dysfunction and provide appropriate interventions.

Our findings have significant implications for clinical practice and research. Future research should aim to further investigate the complex relationship between PCOS and sexual dysfunction.

## Figures and Tables

**Figure 1 healthcare-11-01488-f001:**
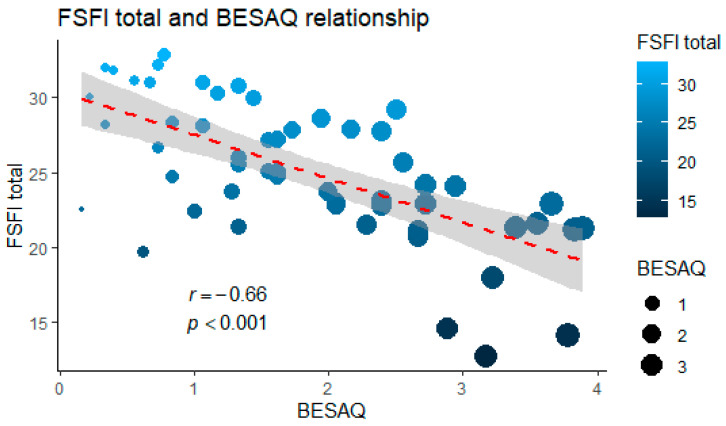
Correlation analysis (each dot on the graph represents a patient included in the study, the size of the dot being given by the BESAQ score, and the color by the FSFI score).

**Figure 2 healthcare-11-01488-f002:**
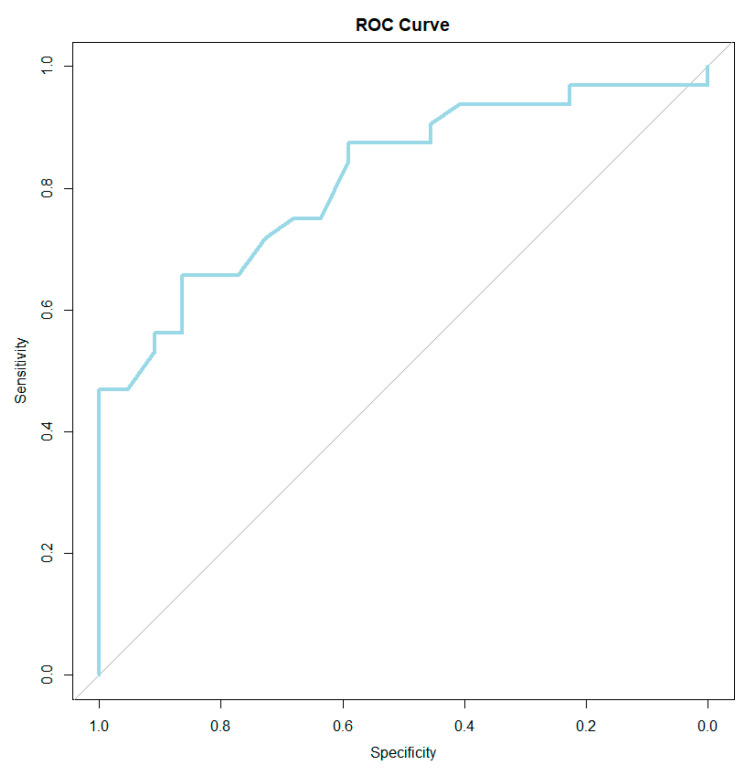
ROC curve for BESAQ score.

**Table 1 healthcare-11-01488-t001:** Baseline characteristics.

Observations = 54	
Living area	
Rural	12 (22.2%)
Urban	42 (77.8%)
Education	
Gymnasium	2 (3.7%)
High school	12 (22.2%)
College	40 (74.1%)
Professional status	
Employed	41 (75.9%)
Unemployed	13 (24.1%)
Civil status	
Married	30 (55.6%)
Unmarried	24 (44.4%)
Religion	
Orthodox	39 (72.2%)
Catholic	9 (16.7%)
Protestant	5 (9.3%)
Agnostic	1 (1.8%)
FSD	
Positive	32 (59.3%)
Negative	22 (40.7%)

Abbreviations: FSD—Female Sexual Dysfunction.

**Table 2 healthcare-11-01488-t002:** Continuous variables within the sample.

Variables	Mean/Median	SD/Q25–Q75
* FG score	16.89	±2.90
* AC	90.83	±15.66
* BESAQ	1.83	±1.04
* Age	26.54	±2.94
BMI	26.55	23.73–35.50
OM	90.00	39.50–148.80
FT	4.89	3.16–5.47
LH/FSH	2.78	2.47–2.96
cortisol	16.65	14.35–23.30

* Normally distributed data; Abbreviations: FG score—Ferriman–Gallwey score; AC—abdominal circumference; BESAQ—Body Exposure during Sexual Activities Questionnaire; SD—standard deviation; BMI—Body mass index; OM—Maximum duration of oligomenorrhea; FT—Free testosterone; and LH/FSH—Luteinizing hormone/Follicle stimulating hormone Ratio.

**Table 3 healthcare-11-01488-t003:** FSFI domains and scores.

Scores	Mean/Median	SD/Q25–Q75
FSFI pain	4.80	4.00–6.00
FSFI desire	3.60	3.00–5.40
FSFI arousal	4.05	3.00–5.10
FSFI lubrication	4.80	3.37–5.40
FSFI orgasm	4.00	2.40–5.20
FSFI satisfaction	4.80	3.30–5.20
* FSFI total	25.08	±4.62

* Normally distributed data; Abbreviations: SD—standard deviation; Q25–75—the first and the third quartile; and FSFI—Female Sexual Function Index.

**Table 4 healthcare-11-01488-t004:** Comparison between normal weight and overweight patients regarding other determinants.

Variable	*p*-Value	Rank Biserial
FT	0.11	−0.26
LH/FSH	0.25	−0.19
BESAQ	0.02	0.39
FSFI total	<0.001	−0.52

Abbreviations: BMI—Body mass index; FT—Free testosterone; LH/FSH—Luteinizing hormone/Follicle stimulating hormone Ratio; FSFI—Female Sexual Function Index; and BESAQ—Body Exposure during Sexual Activities Questionnaire.

**Table 5 healthcare-11-01488-t005:** Correlation analysis between FSFI domains and other determinants.

Correlations	Correlation Coefficient	*p*-Value
FSFI total—BMI	−0.56	<0.001
FSFI total—AC	−0.51	<0.001
FSFI total—BESAQ	−0.66	<0.001
FSFI orgasm—BMI	−0.53	<0.001
FSFI orgasm—cortisol	−0.59	<0.001
FSFI orgasm—AC	−0.52	<0.001
FSFI orgasm—BESAQ	−0.52	<0.001
FSFI arousal—BMI	−0.53	<0.001
FSFI arousal—BESAQ	−0.80	<0.001
FSFI desire—BESAQ	−0.64	<0.001

Abbreviations: FSFI—Female Sexual Function Index; BMI—Body mass index; AC—abdominal circumference; and BESAQ—Body Exposure during Sexual Activities Questionnaire.

**Table 6 healthcare-11-01488-t006:** Logistic regression for determining risk factors.

	Odds Ratios	CI	*p*-Value
BESAQ	4.24	2.04–10.83	0.001

Abbreviations: BESAQ—Body Exposure during Sexual Activities Questionnaire; CI—95% confidence interval.

**Table 7 healthcare-11-01488-t007:** Linear regression for predicting female sexual dysfunction.

	Estimates	CI	*p*-Value
FT	−2.24	−3.22–−1.26	<0.001
BESAQ	−2.26	−3.30–−1.22	<0.001
LH/FSH	3.92	1.18–6.67	0.006
BMI	−0.27	−0.46–−0.09	0.005

Abbreviations: FT—Free testosterone; BESAQ—Body Exposure during Sexual Activities Questionnaire; LH/FSH—Luteinizing hormone/Follicle stimulating hormone Ratio; BMI—Body mass index; and CI—95% confidence interval.

**Table 8 healthcare-11-01488-t008:** ROC curve parameters for BESAQ in relation to diagnosis of FSD.

	BESAQ
AUC	0.816 (0.704–0.928)
Sensitivity	65.6%
Specificity	86.3%
Accuracy	74.1%
*p*-value	<0.001
BESAQ Threshold Value	1.92

Abbreviations: AUC—area under the curve; BESAQ—Body Exposure during Sexual Activities Questionnaire.

## Data Availability

The data presented in this study are available on request from the corresponding author. The data are not publicly available due to reasons concerning privacy of the subjects.

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
