# Peer review of "Effects of Hormonal Profile, Weight, and Body Image on Sexual Function in Women with Polycystic Ovary Syndrome"

_healthcare, 2023, doi:10.3390/healthcare11101488_

Round 1
Reviewer 1 Report
This manuscript presents descriptive and inferential analyses regarding the hormonal, clinical, and psychometric components of PCOS on women's sexual dysfunction. Overall, the manuscript presents data and ideas that are interesting and could contribute to the fields of endocrinology and psychology, or more importantly -- to a more holistic approach to understanding women's health. However, the manuscript is somewhat confusing to read in its current version, and a few recommendations for revision are made to improve clarity.
1. The introduction presents relevant literature regarding the hormonal, clinical, and psychological effects of PCOS on sexual function in women. It is recommended that additional literature is included regarding the psychological component, as this component is not explored as much as the other two components.
The last paragraph of the introduction jumps back to the hormonal component, and indicates the purpose of the study is to identify predictive factors of sexual dysfunction in PCOS patients. However, the purpose of the study should include more than the hormonal component. This paragraph should be revised to include predictions about the clinical and psychological components of PCOS as well. Overall, the purpose of the study should be made more clear and include all variables.
2. The discussion provides summaries and potential explanations for the results found. However, it is somewhat difficult to understand the connections made in this section. One example is this statement:
"Recent studies found that there was no correlation between the BESAQ and the FSFI scores, showing that there is no link between perception of body image and sexual function [48, 49]. However, our results showed that high BESAQ scores are correlated with low FSFI scores, like previous studies suggested [50]." (pp 11)
The manuscript is saying the literature found no correlation, but that the current study did find a correlation ... like the literature. These types of contradictions should be revised and clarified throughout the discussion.
Reviewer 2 Report
This was an interesting manuscript to review.
Addressing the following should improve the quality of the study.
1. The introduction should ideally reduce the narrative providing an overview of PCOS and expand on the background research on sexual dysfunction in PCOS.
2. Please rephrase the following sentence in the third paragraph of the introduction, as I do not understand it. “Numerous studies show that the prevalence rates of PCOS in accordance with obesity, however, are close for the categories of underweight, normal- weight, overweight women with mild, moderate and severe obesity, respectively 8.2%, 9.8%, 9.9 %, 5.2%, 12.4% and 11.4%.”
3.Clarify the hypothesis aims and objectives.
4. Comment on any sample size considerations before the study. If this was not done, please justify the decision.
5. What was the primary outcome measure?
6. Would it have helped to also have BMI matched controls? If not, please justify. This might have been useful to exclude the possible confounding effects of obesity.
7. Is it possible to provide data on how the length of amenorrhoea correlated with female sexual dysfunction?
8. Please state the source of patient recruited into the study.
9. Please include the BMI thresholds used to define normal weight, and overweight in table 4, and the numbers in each group.
10. It would have been helpful to collect data on the mental wellbeing of the study participants (e.g., depression and anxiety) and investigated the independent and confounding effect of these variables on the diagnosis of Female Sexual Dysfunction in PCOS. Is it possible to do this? If not, consider including this in the discussion, as a limitation of the study.
11. It would have been good to measure oestrogen levels (e.g., oestradiol) investigated its independent on the diagnosis of Female Sexual Dysfunction in PCOS. If this was not considered necessary, please justify this in the discussion.
12. The 1st paragraph of discussion should go into introduction.
13. Please rephrase the following sentence in paragraph 3 on your discussion “Therefore, we observe a U- shaped effect of testosterone because both testosterone deficiency and excess, as in the case of PCOS, affect sexual function”, as it could be interpreted to mean that you provided data in your results to support this assertion, and I don’t recall seeing data to support this, in your results section.
14. In the conclusion, please expand further on the implications of your study for clinical practice and research. What does all this mean?
English language and grammar review required.
The article needs some review of the English language and grammer.
